# Physical Activity Frequency and Health-Related Quality of Life in Spanish Children and Adolescents with Asthma: A Cross-Sectional Study

**DOI:** 10.3390/ijerph192114611

**Published:** 2022-11-07

**Authors:** Ángel Denche-Zamorano, Raquel Pastor-Cisneros, Lara Moreno-Moreno, Jorge Carlos-Vivas, María Mendoza-Muñoz, Nicolás Contreras-Barraza, Miseldra Gil-Marín, Sabina Barrios-Fernández

**Affiliations:** 1Health, Economy, Motricity and Education Research Group (HEME), Faculty of Sport Sciences, University of Extremadura, 10003 Caceres, Spain; 2Social Impact and Innovation in Health (InHEALTH), University of Extremadura, 10003 Caceres, Spain; 3Promoting a Healthy Society Research Group (PHeSO), Faculty of Sport Sciences, University of Extremadura, 10003 Caceres, Spain; 4Research Group on Physical and Health Literacy and Health-Related Quality of Life (PHYQOL), Faculty of Sport Sciences, University of Extremadura, 10003 Caceres, Spain; 5Departamento de Desporto e Saúde, Escola de Saúde e Desenvolvimento Humano, Universidade de Évora, 7004-516 Evora, Portugal; 6Facultad de Economía y Negocios, Universidad Andres Bello, Vina del Mar 2531015, Chile; 7Public Policy Observatory, Universidad Autónoma de Chile, Santiago 7500912, Chile

**Keywords:** asthma, health-related quality of life, physical activity

## Abstract

Asthma is considered the most prevalent chronic childhood disease worldwide. Physical activity (PA) represents a tool to improve patients with respiratory diseases’ health-related quality of life (HRQoL). A cross-sectional study examining data from the Spanish National Health Survey (ENSE 2017) was carried out to investigate the associations between the PAF (physical activity frequency) and health-related quality of life (HRQoL) in asthmatic children and adolescents from 8 to 14 years old (total 11.29 years ± 1.91, boys 11.28 ± 1.90 and 11.29 ± 1.95 girls). Data were obtained from the Children Questionnaire, which was completed by their families or legal guardians, achieving a final sample composed of 240 participants with asthma. Data were taken from Survey 2017 (ENSE 2017), the last one before the COVID-19 pandemic. The results revealed significant associations between the PAF and the HRQoL, finding no significant differences between genders.

## 1. Introduction

Asthma is a chronic inflammatory airway disease characterised by bronchoconstriction, airway inflammation and mucus hypersecretion and leading to variable airflow limitation, including attacks with diverse severity and frequency with bronchial hyperresponsiveness [1,2]. This condition usually causes wheezing, a whistling sound on expiration, and in severe cases, also during the inspiration phase. Asthma also can cause shortness of breath, chest tightness and cough, especially in the child population [3]. Asthma is considered a major cause of disability and reduced quality of life and death in children and is the most prevalent chronic childhood disease in the world [3,4,5]. Although the reported prevalence has been higher in developed countries, a high prevalence of severe symptomatology is detected in low- and middle-income countries [6,7]. In Spain, this disease management is costly and causes a high social impact, as it involves a large number of direct, indirect and intangible costs [8]: asthma is the most frequent cause of hospital admission and visits to the emergency department [9]. The severity of the pathology is the main predictor of cost. In Spain, one study estimated that the cost of asthma in the pediatric population could reach 532 million euros, with a cost per child ranging from EUR 403 to 5380 [10]. Risk factors related to childhood asthma, both genetic and environmental, have been studied in different countries around the world, based on their development in urban and rural areas. Risk factors studied include a family history of asthma, prematurity and low birth weight, exposure to tobacco smoke both during pregnancy and after birth, allergies, recurrent respiratory infections, living in polluted areas and obesity, among others [4,11,12].

According to the World Health Organization (WHO), health-related quality of life (HRQoL) is defined as “an individual’s perception of their position in life in the context of the culture and value system in which they live and concerning their goals, expectations, norms and concerns” [13]. HRQoL measures are subjective (reported as the perception of the person involved), multidimensional (reporting on various aspects of the individual’s life: physical, emotional, social, interpersonal, etc.) and include positive and negative feelings as well as varying over time (age, stage of life, time of illness) [14]. A decrease in HRQoL results in the child having less chance of developing and maturing normally, which may mean that the child does not grow up to be a healthy adult. A study suggests that improved cardiorespiratory fitness may influence the symptoms of depression and self-esteem, leading to enhanced well-being [15]. Currently, low levels of HRQoL are more common when body mass index (BMI) levels exceed WHO recommended levels [16], and some studies have found that individuals with asthma have a statistically significant higher BMI than healthy people [1,17,18]. As a result, decreased HRQoL is observed in children and adolescents with asthma [19], and children with severe asthma show even worse HRQoL and behavioural issues [20].

Asthma may reduce the perceived ability to participate in PA, which may contribute to increased sedentary lifestyles [21]. Negative self-efficacy, parental beliefs about PA and children’s asthma control or fear of an asthma attack could be barriers to an active lifestyle in children with asthma. This, together with a lack of opportunities to access PA programs, combined with easy access to mobile and tablet technologies, video games and social media, represent a contributor to the increase in sedentary lifestyles and physical inactivity in this population [22,23]. Some studies have shown that children with asthma have a higher prevalence of inactivity than the general population of their age, being higher in girls than in boys [24,25]. However, PA is a valuable resource as it helps in increased cardiopulmonary capacity, asthma symptomatology and HRQoL [9,17,21,26]. Moreover, some studies suggest that improved cardiorespiratory fitness may improve mood and self-esteem [15]. Furthermore, PA may induce anti-inflammatory effects [27]. Some of the main exercise modalities recommended are aerobic training (high-intensity interval training, cycle ergometer, treadmill) [28], and strength [29]. Additionally, swimming favours lung mucosa mobilization and expulsion in humid and warm environments and the horizontal position [30].

PA frequency (PAF) is important when studying the benefits for children and adolescents diagnosed with asthma. Several studies have shown the association between PA and asthma: lower PA was inversely related to asthma [31], with the direction of the associations turning positive as weekly PA increased [25]. Thus, this study aims to evaluate potential associations between the PAF and HRQOL in Spanish children and adolescents aged between 8 and 14 years diagnosed with asthma, also searching for differences between sex.

## 2. Materials and Methods

### 2.1. Study Design and Ethical Concerns

A cross-sectional study was conducted using data from the Children Questionnaire administered for ENSE 2017 [32] This survey is administered by the Spanish Ministry of Health, Consumption and Social Welfare and the National Institute of Statistics every five years to assess health-related factors to improve health policies. Specifically, the Children Questionnaire [33], which contains information on sociodemographic variables and information on the state of health, is divided into three modules: health condition, health determinants and health care.

According to Regulation 2016/679 of the European Parliament and of the Council of 27 April 2016 on the protection of individuals about the processing of personal data and the free movement of personal data and derogating from Directive 95/46/EC [34], anonymous files for public use are not considered confidential. Thus, approval by a bioethics committee is not required.

### 2.2. Participants

The sample used was extracted from the Spanish National Health Survey (ENSE 2017), through the Children Questionnaire. The eligibility criteria included (1) individuals between 8 and 14 years (2) who had an asthma diagnosis; (3) those participants who scored outside the 1–5 range on “block E” related to HRQoL and (4) those who scored outside the 1–4 range on “block K”, related to PA habits were excluded. Thus, although the initial sample size was 6106 participants, after applying these criteria, the final sample was composed of 240 participants (Figure 1).

### 2.3. Procedure and Variables

The variables were collected and calculated through the Children Questionnaire, using the following blocks: “Block B: Health status and chronic morbidity”: used to identify participants with asthma (question 2, value 1).“Block E: Quality of life:” using the Kidscreen-10 Index modified proxy for the Eurobarometer, composed of 9 items instead of 10 (KS9) [35,36]. The response options were (1) “not at all”, (2) “a little”, (3) “moderately, (4) “a lot” and (5) “very much”. It was constructed, with the sum of items E14.1-E14.9 (inverting the values of items: E14.3 and E14.4), divided by the number of items, and can take values between 1 (worst HRQoL) and 5 (best HRQoL).“Block K: Resting and PA”: question 61 was considered, as it was the only one referring to leisure-time PAF. This question offered four response options: (1) “no exercise: free time is spent mostly in a sedentary way (reading, watching TV, going to the cinema, etc.)”; (2) “Occasionally does some PA or sport (walking or cycling, gentle gymnastics, recreational activities requiring light effort, etc.); (3) “Does PA several times a month (sports, gymnastics, jogging, swimming, cycling, team games, etc.)”; (4) “does sports or physical training several times a week”.

### 2.4. Statistical Analysis

Statistical analyses were performed using the Statistical Package for Social Sciences 25 (IBM SPSS, Armonk, NY, USA). The Kolmogorov-mirnov test was used to know the data distribution for the different variables. A descriptive analysis was performed using the median and the interquartile range, with mean and standard deviation (continuous variables) and absolute and relative frequencies (ordinal variables). The Mann–Whitney U test for two independent samples was used to determine differences between genders. Kruskal–Wallis’s test was carried out to check the differences in the HRQoL levels according to PAF using PA values as factors. A post hoc analysis for multiple comparisons, using the Bonferroni correction (*p* < 0.01) was performed. The Mann–Whitney U test for two independent samples was used to determine whether differences existed between PAF, grouped as those who do no or little PA and those who train several times a week. Finally, a bivariate correlation to determine the association between HRQoL and PAF with Spearman’s correlation coefficient was performed.

## 3. Results

Table 1 shows the characterization of the total sample and grouped by sex, including information about their PAF, and reported HRQoL.

Table 2 shows the results obtained when comparing the HRQoL with the PAF, showing that the population with the highest PAF presented the highest HRQoL, with significant differences between the “several times a week” PA group and the “occasional” or “several times a month” PA groups (*p* < 0.05 and *p*< 0.001, respectively). 

There were significant differences between groups according to their PAF (very active and inactive/occasionally/several times a month) concerning their HRQoL. Children and adolescents with asthma who performed little or no PA reported worse HRQoL compared with those who engaged in PA several times a week (Table 3).

Table 4 shows the correlations between the HRQoL level and the PAF.

## 4. Discussion

### 4.1. Main Findings and Theoretical Implications

The main finding of this research is the significant association found between PAQ and HRQoL in Spanish children and adolescents aged 8–14 years with asthma. Several studies on the benefits of PA in HRQoL in children and adolescents with asthma [37,38,39,40] have shown that an increased PAF improves HRQoL [25]. There are other reported health benefits such as a lower risk of suffering other chronic diseases [9,17,38,39,41,42], better lung capacity [9,17], and mood enhancement [9,17,38]. Moreover, regular PA practice protects against comorbidities such as anxiety, depression and obesity, [38,39,42,43]. In this regard, our results revealed significant differences in HRQoL between those engaged in regular PA and those less active or inactive, in line with previous research [37,40,44,45]. One of the reasons why the literature indicates that PA could improve HRQOL in children and adolescents with asthma is that it could help in the control of the symptoms of this pathology due to cardiorespiratory adaptations to exercise [9], body mass index decrease [1,41,45] self-esteem and mood improvement [41] or the reduction of symptoms related to anxiety and depression [15,43].

The tool the Children Questionnaire from the ENSE 2017 was the Kidscreen-10 Index modified proxy for the Eurobarometer. When correlating its items with the PAF, only item eight (“Has your child got on well at school?”) was statistically significant. This item encompasses both academic and social factors that children put into practice in the school environment. Thus, on the one hand, previous studies have found relationships between the level of PA and academic performance, with more active children and adolescents obtaining better grades, and on the other hand, the practice of PA could produce emotional and social improvements as it can help or strengthen relationships of friendship and cooperation [46]. In fact, in our results, items four (“Has your child felt lonely?”) and seven (“Has your child had fun with their friends?”), although correlated with PAF, were not significant. This, again, could be explained by the small sample size.

Although no statistically significant gender differences between the PAF and HRQoL were found (Table 1), several studies have reported significant differences [43,47,48,49]. This could be explained by the small sample size, as we found average PA values slightly higher in boys than in girls in line with the above-mentioned studies. 

### 4.2. Practical Implications

The strength of this study lies in the analysis of associations between PAF and HRQoL in Spanish asthmatic children and adolescents during the last period before the COVID-19 pandemic, which could serve as a frame of reference for future research examining post-pandemic periods, as the ENSE is addressed every 5 years. In addition, because of the negative consequences caused by the pandemic on the daily habits of the Spanish population in terms of physical activity and sedentary behaviour, the data from this study can serve as a comparative framework to study the post-pandemic situation, with the next survey expected to be completed in 2023.

The associations found showed that there was a significant association between PA and HRQOL in children and adolescents aged 8–14 years with asthma. Although several studies recommend the implementation of PA programs [22,23] or “sports prescription” as a cost-effective alternative to reduce health expenditure, the design of our study did not allow us to establish cause-effect associations. 

This research topic can be useful to physical education and other stakeholders working with children and adolescents with asthma diagnosis, reinforcing the available recommendations for this disease’s management [50]. However, professionals should the prepared to deal with this disease because although a great number of studies support PA’s positive effects on physical and mental health both in pediatric [51] and asthmatic populations [31], further research should clarify the relationships between the PA level/load and children with asthma characteristics/symptomatology to improve the quality of the interventions to promote the health and quality of life in children and adolescents with asthma [24,52,53].

### 4.3. Limitations and Future Lines

This study has some limitations. Subjects reported being asthmatic without evidence of a medical diagnosis, which also results in a lack of knowledge of the specific asthma symptoms experienced by the participants: the lack of participants’ medical histories, PA objectives, and physiological data, including follow-ups, are aspects that could be improved. Among others, implementing a 24 h compositional analysis, with devices to quantify physical activity intensity or other measures may help to overcome some of the limitations of survey-based studies. Moreover, the ENSE 2017 Children Questionnaire does not include questions related to important aspects such as motivation or self-esteem [54], as well as sociocultural factors such as economic status, which are recommended to be included in subsequent editions and future studies. In addition, it would be interesting to know what happens in the transition from primary to secondary school as at this stage; the dropout from sport is imminent and could therefore affect the severity of asthma. Furthermore, the study design did not allow for establishing cause-effect relationships, so future research should include longitudinal studies.

## 5. Conclusions

Children and adolescents with asthma with higher PAF show higher HRQoL scores compared with those with lower PAF. No statistically significant gender differences between the PAF and HRQoL were found. 

## Figures and Tables

**Figure 1 ijerph-19-14611-f001:**
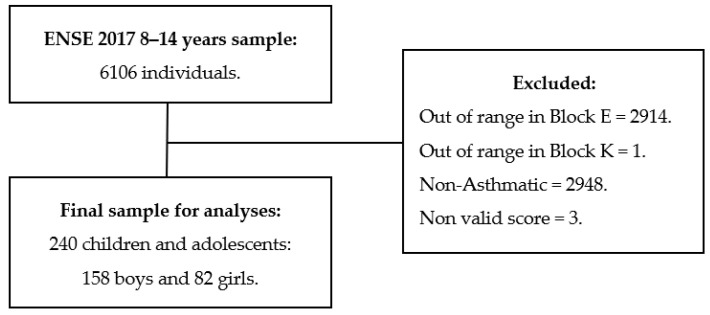
The study sample’s eligibility criteria process.

**Table 1 ijerph-19-14611-t001:** Sociodemographic characteristics of the Spanish children and adolescents with Asthma participants.

	Total	Boys	Girls	*p*
Age (years)	8–14	8–14	8–14	
Number	24	158	82	
Median (IQR)	11 (3)	11 (3)	11 (3)	0.98
Mean (SD)	11.29 (1.91)	11.28 (1.9)	11.29 (1.95)	
PAF				
Median (IQR)	3 (2)	3 (2)	3 (2)	0.44
Mean (SD)	2.95 (1.04)	2.99 (1.03)	2.88 (1.06)	
HRQoL				
Median (IQR)	3.78 (0.44)	3.78 (0.44)	3.78 (0.33)	0.78
Mean (SD)	3.68 (0.35)	3.68 (0.36)	3.69 (0.33)	

IQR: interquartile range; SD: standard deviation; *p*-value: Mann-Whitney’s U test; PAF: physical activity frequency (values 1–4); HRQoL: Health-Relates Quality of Life (values 1–5).

**Table 2 ijerph-19-14611-t002:** Health-related quality of life comparison with physical activity frequency in spanish children and adolescents with asthma.

PAF		HRQoL	PAF	Means Diff.	Medians Diff.	ρ *	ρ **
1			2	0.08	0.17	0.010	0.429
	Median (IQR)	3.78 (0.50)	3	0.06	0.11		0.184
	Mean (SD)	3.68 (0.40)	4	−0.10	0.00		0.323
2			1	−0.08	−0.17	0.010	0.429
	Median (IQR)	3.61 (0.64)	3	−0.02	−0.06		0.991
	Mean (SD)	3.60 (0.42)	4	−0.18	−0.17		0.036
3			1	−0.06	−0.11	0.010	0.184
	Median (IQR)	3.67 (0.33)	2	0.02	0.06		0.991
	Mean (SD)	3.61 (0.31)	4	−0.16	−0.11		0.001
4			1	0.10	0.00	0.010	0.323
	Median (IQR)	3.78 (0.44)	2	0.18	0.17		0.036
	Mean (SD)	3.77 (0.31)	3	0.16	0.11		0.001

IQR: interquartile range. SD: standard deviation. PAF: physical activity frequency (values 1–4). HRQoL: Health-Related Quality of Life (values 1–5). Means Diff.: means differences. Medians Diff.: mediants differences. * Kruskal-Wallis’s test. ** Mann-Whitney’s U test, using the Bonferroni correction (*p* < 0.01).

**Table 3 ijerph-19-14611-t003:** Health-related quality of life and physical activity frequency comparison in Spanish children and adolescents with asthma.

	PAF Levels 1, 2 and 3 (*n* = 150)	PAF level 4 (*n* = 90)	*p* *
Median (IQR)	3.66 (0.44)	3.78 (0.44)	0.002
Mean (SD)	3.62 (0.36)	3.77 (0.31)	

PAF: physical activity frequency; PAF levels 1, 2 and 3: perform little or no physical activity; PAF level 4: does sports or physical training several times a week; *p*: *p*-value from Mann-Whitney’s U test for two independent samples using the Bonferroni correction (*p* < 0.01); *p* *: statistically significant correlation.

**Table 4 ijerph-19-14611-t004:** Kidscreen-10 Index modified proxy for the Eurobarometer and the physical activity frequency in Spanish children and adolescents with asthma.

Items	ρ	*p*
Has your child felt fit and well?	0.107	0.094
2. Has your child felt full of energy?	0.100	0.012
3. Has your child felt sad?	0.029	0.650
4. Has your child felt lonely?	0.127	0.050
5. Has your child had enough time for themselves?	0.114	0.077
6. Has your child been able to do the things they want to do in their free time?	0.026	0.685
7. Has your child had fun with their friends?	0.147	0.023
8. Has your child got on well at school?	0.196	0.002
9. Has your child been able to pay attention?	0.116	0.073

ρ: Spearman’s correlation. *p:* the correlation is significant after applying the Bonferroni correction *p* < 0.01.

## Data Availability

Data used were obtained from public use files, available on the Spanish Ministry of Health, Consumer Affairs, and Social Welfare website: https://www.mscbs.gob.es/estadEstudios/estadisticas/encuestaNacional/encuesta2017.htm (accessed on: 15 July 2022). Additional datasets will be available under reasonable request.

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
