# Peer review of "Physical Activity Frequency and Health-Related Quality of Life in Spanish Children and Adolescents with Asthma: A Cross-Sectional Study"

_ijerph, 2022, doi:10.3390/ijerph192114611_

Round 1
Reviewer 1 Report
You have written an interesting manuscript about physical activity's effects on an age cohort of Spanish children with asthma. The importance of this work is self-evident. However, there are many ways to improve the text. I will comment on each section separately.
Abstract
Line 28: the average age of the girls in the sample is missing
Line 30-31: The sentence that begins with "Survey 2017" is an incomplete sentence. You can say "Data are taken from Survey 2017 (ENSE 2017), the last one..."
Line 31-34: I recommend removing this sentence. You do not need to mention statistical methods in the abstract.
Introduction
Line 53: Where is asthma the most frequent cause of hospital admission? In Spain, or globally?
Line 77-79: Re-write this sentence to "As a result, decreased HRQoL is observed in children and adolescents with asthma, and children with severe asthma show even worse HRQoL and behavioural issues."
Line 89-91: Cardiovascular and aeroabic exercise are the same thing. Please re-write this sentence.
Line 94, and many other times in the manuscript: the word Asthma is capitalized. It should not be, unless it is at the start of a sentence. It is ok to say "asthma".
Line 98-99: It is unnecessary to say "in a pre-pandemic period". Please delete.
Materials and Methods
You use Mann-Whitney and Mann-Whitney's interchangeably. Please choose one format and stick to it. Same with Kruskal-Wallis.
Line 146: This should be a period and a new sentence, not a semicolon. "...differences between sexes. The Kruskal-Wallis..."
Results
Line 162-166: My understanding of the Results section is that you do not need to elaborate on the statistical methodology that was used. It is better to recite the results, say what they showed, and what the statistical significance of these findings was. You have already discussed statistical tools in the Methods section. Same for lines 183-185, in which you mention Spearman's correlation. You can mention the result without talking about the statistical tool.
Line 175: Delete the word "exists". Write "There were significant differences between the two groups..."
Discussion
Line 210: Do you mean to say Table 3, not Table B3?
Line 220-224: These sentences are poorly written and it is not clear to me what you are trying to claim. Please consider re-writing.
Overall I think section 4.1 needs a lot of work. Nowhere do you discuss whether the magnitude of a 0.1 difference in HRQoL between asthma patients who engage in PA frequently and those who do not is truly meaningful. What is the scale of the HRQoL? This is also worth mentioning in the discussion.
Section 4.2 and Section 4.3 are well-done, as is the Conclusion section.
Author Response
Dear Editor,
You will find the authors' reply in the attached file.

Reviewer 2 Report
The importance of this study lies in the analysis of associations between physical activity frequency and HRQoL in Spanish asthmatic children and adolescents during the last period before the COVID-19 pandemic.
This study has some limitations that should be taken into account when interpreting the results and drawing conclusionsbut the statistical power of the sample included in the study is 88%.
Based on the results obtained, there is a statistically significant direct correlation between the PAF and HRQOL in children and adolescents aged 8–14 years with asthma. Future studies are also needed to investigate how PA improves HRQoL in the asthmatic population.
Author Response

(The authors gave the same response as above.)

Reviewer 3 Report
Reading and reviewing the article has been a highly interesting task for me. From my background in research, measuring the levels of Physical Activity in groups with disabilities or diseases and in institutionalized groups is an essential task to achieve active and healthy societies.
Apart from this I leave below some considerations related to the article:
Introduction:
Line 71: the authors refer to "some Studies" but only cite one.
Lines 81-85: at this point they try to see the relationship between PA, sedentary lifestyle and asthma. But I fail to see the relationship between the three. I think it is necessary to better explain the relationship between PA and asthma. Conditions that lead to sedentary lifestyles in children with asthma. Recommendations for this type of people. In the same way the approach to PA among boys and girls is different (in the objective it is marked as a research question).
Discussion:
Line 204: use the term Gender instead of sex.
Line 206: the similarity between genders cannot only be supported by the size of the sample. It should be discussed that the approach to PA between genders is different at school ages.
Line 208: is there evidence that there are no significant positive differences between those who practice PA and those who do not?
Line 229: Motivation and self-steem as variables to be studied are not analyzed and should be discussed in more detail.
Conclusions
It would be necessary to see if the stated objective is correct and can be assumed. On the other hand, the conclusion of the need to prescribe PA to children and young people with asthma with the help of a professional who understands their needs when practicing.
Author Response

(The authors gave the same response as above.)

Round 2
Reviewer 1 Report
This is a very improved revision. Well done and congratulations on your important research findings.